# Dynamics of type IV collagen 7S fragment on eradication of HCV with direct antiviral agents: Prognostic and metabolomic impacts

**Karin Yamataka[1], Po-sung Chu[1]\*, Yuzo Koda[1,2], Nobuhito Taniki[1], Rei Morikawa[1], Aya Yoshida[1], Fumie Noguchi[1], Ryosuke Kasuga[1], Takaya Tabuchi[1], Hirotoshi Ebinuma[1,3], Takanori Kanai[1], Nobuhiro Nakamoto[1]\***

**1** Division of Gastroenterology and Hepatology, Department of Internal Medicine, Keio University School of Medicine, Shinjuku-ku, Tokyo, Japan, **2** Research Unit/Immunology & Inflammation, Sohyaku Innovative Research Division, Mitsubishi Tanabe Pharma Corporation, Kanagawa, Japan, **3** Department of Gastroenterology, International University of Health and Welfare, School of Medicine, Narita City, Chiba, Japan

\* pschu0928@iCloud.com (PC); nobuhiro@z2.keio.jp (NN)

**Data Availability Statement:** All relevant data are within the paper and its Supporting information files.

## Abstract

### Background

Liver fibrosis is one of the cardinal clinical features of chronic hepatitis C (CHC). However, the mechanisms underlying the evolution and reversion of liver fibrosis after hepatitis C virus (HCV) eradication and their relationship with clinical outcomes and metabolic alterations are not fully elucidated. Whether any non-invasive fibrosis marker can predict prognosis is unknown.

### Methods

Between October 2014 and September 2019, 418 patients with CHC or compensated cirrhosis with HCV were prospectively recruited in this observational study. 326 patients that were successfully eradicated with interferon-free direct antiviral agents (IFN-free DAAs) were analyzed. Peri-treatment dynamics of serum levels of type IV collagen 7S fragment (4COL7S), a fibrosis marker, and subsequent clinical outcomes, including hepatic decompensation, newly emerged hepatocellular carcinoma (HCC), and all-cause mortality were analyzed.

### Results

Ten (3.1%) patients died during the observation period. 4COL7S-defined fibrosis progression (n = 97, 29.8%) at SVR was significantly correlated with worse all-cause mortality post-SVR ($P = 0.0062$) but not with the probability of newly emerged HCC ($P = 0.24$). Prognostic tendency was more prominent in patients with advanced fibrosis ($P < 0.0001$). 4COL7S-defined fibrosis progression at SVR and a baseline platelet count less than $10 \times 10^4/\mu L$ were significantly predicted all-cause mortality ($P = 0.0051$). In exploratory analyses, a decreased 4COL7S at the end of treatment was correlated with a matrix-degrading phenotype that

**Funding:** This study was supported by grants-in-aid from the Keio Gijuku Academic Development Funds and the Mitsubishi Tanabe Pharma Corporation. The funders had no role in study design, data collection and analysis, decision to publish, or preparation of the manuscript. There was no additional external funding received for this study.

**Competing interests:** Y.K. is an employee at Mitsubishi Tanabe Pharma Corporation. The remaining authors declare no competing interests. This does not alter our adherence to PLOS ONE policies on sharing data and materials.

**Abbreviations:** 4COL7S, type IV collagen fragment 7S; AUROC, area under the receiver operating characteristic; CE-TOFMS, capillary electrophoresis time-of-flight mass spectrometry; CHC, chronic hepatitis C; CLD, chronic liver disease; DAA, direct antiviral agent; ECM, extracellular matrix; eGFR, estimated glomerular filtration rate; EOT, end of treatment; GPC, glycerophosphocholine; HCA, hierarchical cluster analysis; HCC, hepatocellular carcinoma; HCV, hepatitis C virus; HSC, hepatic stellate cell; IFN, interferon; MMP, metalloproteinase; MT, migration time; NAFLD, non-alcoholic fatty liver disease; PCA, principal component analysis; PLS-DA, discriminant analysis of partial least squares; Pre, pre-treatment; RECICL, Response Evaluation Criteria in Cancer of the Liver; SVR, sustained virological response; T2DM, type 2 diabetes mellitus; TCA, taurocholic acid; TIMP, tissue inhibitor of metalloproteinase.

showed higher serum metalloproteinase to tissue inhibitors of metalloproteinase-1 ratios and characteristic metabolic fingerprints such as increased butyrate, some medium-chain fatty acids, anabolic amino acids, and decreased uremia toxins.

## Conclusions

Peri-treatment dynamics of serum 4COL7S, a non-invasive fibrosis marker, predict prognosis. Non-invasive fibrosis markers may be useful biomarkers for risk stratification post-SVR.

## Introduction

Chronic liver disease (CLD) is a major cause of mortality and morbidity worldwide [1]. One of the major etiologies of CLD is the hepatitis C virus (HCV), which is estimated to persistently infect approximately 1% of the global population [2]. The incidence remains high despite the clinical utilization of the paradigm-changing interferon (IFN)-free direct antiviral agents (DAAs), which result in over 90% sustained virological response (SVR) rates.

In patients with chronic hepatitis C (CHC), the chronological effect on the progression of liver fibrosis involves the cardinal clinical feature [3] that may lead to cirrhosis, and may be associated with portal hypertension, hepatic decompensation, and hepatocellular carcinoma (HCC) [4]. In addition to liver-related complications, extrahepatic ones, such as non-liver malignancies and metabolic, cardiovascular, and immune-mediated manifestations related to persistent HCV infection have also been reported [5–7]. Reports exploring the impact of anti-HCV treatment, including both IFN-based or IFN-free DAAs and subsequent SVR on clinical outcomes, are accumulating. However, the evidence levels supporting each beneficial effect vary [8]. For instance, liver fibrosis regression has been observed in some, but not all, patients after HCV eradication [9–12]. Moreover, patient characteristics, degree of regression, and clinical outcomes of significant fibrosis regression after SVR remain unknown. In addition, the gold-standard for evaluating fibrosis resolution remains unestablished [13]. Hence, a non-invasive biomarker that predicts clinical outcomes of patients undergoing HCV eradication is needed.

CHC is also a prototype for studying general mechanisms of host-microorganism interactions, disease progression, and alterations in multiple signaling and metabolic pathways [14], especially pathways leading to prominent steatosis with hypermetabolism in hepatocytes [15]. Nevertheless, it remains unclear whether fibrosis reversion is associated with metabolic changes after HCV eradication.

The extracellular matrix (ECM) is dynamic and complex in terms of both quantity and quality during liver fibrogenesis and fibrolysis [13]. During ECM remodeling, metalloproteinases (MMPs), including gelatinase A (MMP2) and gelatinase B (MMP9), inhibitors such as the tissue inhibitor of metalloproteinase-1 (TIMP-1), and their balances have been studied [16]. During hepatic stellate cell (HSC) activation accompanying liver injury, type IV collagen deposition becomes prominent, such that a real basement membrane develops in the perisinusoidal space. This process is called sinusoidal capillarization [17]. Moreover, quantification of the serum 7S fragment of type IV collagen (4COL7S), the amino-terminal triple-helix domain of type IV collagen, is a biomarker for ECM remodeling during liver fibrogenesis [18]. In addition, it has demonstrated better sensitivity and specificity for the detection of cirrhosis due to viral hepatitis compared with serum levels of type IV collagen [19]. Whether any non-invasive fibrosis marker can predict prognosis after SVR is still incompletely studied.

In this observational study of patients with CHC whose HCV was eradicated using IFN-free DAAs, we focused on the peri-treatment dynamics of serum 4COL7S at three time points: immediately before treatment, at the end of treatment, and at 12 weeks after achieving SVR. We also determined their association with the patients' clinical outcomes. The correlation between matrix remodeling and metabolomic changes was also explored.

## Materials and methods

### Study subjects, ethics, and follow-up

This observational study was approved by the Institutional Review Board of Keio University School of Medicine (No. 20140177). This study was conducted in accordance with the 2013 revision of the guidelines of the 1975 Declaration of Helsinki. Patients with CHC before treatment with DAAs were recruited between October 2014 and September 2019. The recruited study subjects provided prior written informed consent to use and publish the results of their blood samples and analysis of clinical data. No study participants aged < 18 years were included. All study subjects received standard care and treatment based on their clinical presentation. Plasma *HCV RNA* was quantified using commercially available kits of COBAS Taqman® (Roche, Switzerland; with a lower limit of quantification of 15 IU/mL). Negative plasma *HCV RNA* at 12 weeks after the end of treatment was considered to achieve SVR (SVR12). Since most of the recruitment period of this study was before an on-label use of sofosbuvir/velpatasvir, and due to the exclusion of national health-care insurance reimbursement to DAAs other than sofosbuvir/velpatasvir for patients with decompensated cirrhosis at baseline, none of such patients was included in this study. For patients with a history of HCC, IFN-free DAAs were started after confirming that all HCC target lesions were completely treated according to Response Evaluation Criteria in Cancer of the Liver (RECICL) 2015. The exclusion criteria were as follows: (i) non-SVR patients; (ii) death within six months after achieving SVR12; (iii) insufficient (less than six months after SVR12) follow-up duration; and (iv) incomplete data for analysis. The regimens of IFN-free DAAs included daclatasvir/asunaprevir for 24 weeks (n = 62), sofosbuvir/ledipasvir (n = 147) for 12 weeks, sofosbuvir/ribavirin (n = 62) for 12 weeks, elbasvir/grazoprevir (n = 13) for 12 weeks, ombitasvir/paritaprevir/ritonavir (n = 14) for 12 weeks, and glecaprevir/pibrentasvir (n = 28) for 8–12 weeks. During treatment and follow-up, all patients underwent routine laboratory examinations at every visit and standard HCC surveillance according to national recommendations [20] every three to six months, including liver ultrasonography or contrast-enhanced computed tomography or magnetic resonance imaging, and tumor markers. All-cause mortality, admission, or liver transplantation due to hepatic decompensation, and novel HCC emergence, including both first occurrence and recurrence, were the predesignated outcomes of interest. The schema of the blood sampling and clinical observation of outcomes are summarized in S1A Fig in S1 File. The inclusion and exclusion flow of this study is summarized in S1B Fig in S1 File.

### Biochemical studies and liver fibrosis indices

Serum 4COL7S levels were measured using the radioimmunoassay two-antibody method (Lumipulse Presto®; Fujirebio Inc., Tokyo, Japan; with a lower limit of quantification of 0.270 ng/mL; reference level for normal adults, < 6.0 ng/mL). This measurement was clinically approved and available using this commercial kit in Japan, and the clinical applications of serum 4COL7S levels derived clinically have been reported in other clinical observations of various acute or chronic liver diseases [21–23]. Non-invasive fibrosis scores, which included aspartate aminotransferase (AST) to platelet ratio index (APRI, cutoff for METAVIR F4 cirrhosis, > 2.0), FIB-4 (cutoff for METAVIR F3-F4 advanced fibrosis, > 3.25), and Forns index

(cutoff for METAVIR F0-F1, < 4.2), were calculated as described previously [24–26]. In the exploratory sub-cohort analysis 1, 23 patients who achieved SVR by taking IFN-free DAAs were included. Sera collected immediately before (Pre) and at end-of-treatment (EOT) were analyzed for the levels of MMP-2 (R&D Systems, Minneapolis, MN, USA), MMP-9 (R&D Systems), and TIMP-1 (Enzo Biochem, New York, NY, USA) via enzyme-linked immunosorbent assays (ELISA).

## Metabolome analysis and data processing

In the exploratory nested case-control analysis 2, seven patients with 4COL7S-defined fibrosis progression at EOT and seven background characteristic-matched patients with fibrosis regression were included. All 14 patients were retrieved from the main cohort, and SVR was achieved. Paired serum samples collected at Pre and EOT were stored at -80˚C since collection until the start of analysis. Approximately 50 μL of serum was added to 200 μL of methanol containing internal standards (H3304-1002, Human Metabolome Technologies, Inc., HMT, Tsuruoka, Yamagata, Japan) at 0˚C to suppress enzymatic activity. The extract solution was thoroughly mixed with 150 μL of Milli-Q water, after which 300 μL of the mixture was centrifugally filtered through a Millipore 5-kDa cutoff filter (ULTRAFREE MC PLHCC, HMT) at 9,100 ×g and 4˚C for 120 min to remove the macromolecules. The filtrate was then evaporated to dryness under vacuum and reconstituted using 50 μL of Milli-Q water for the subsequent metabolome analysis at HMT. Metabolome analysis was conducted according to HMT's Basic Scan package, which required the usage of capillary electrophoresis time-of-flight mass spectrometry (CE-TOFMS) based on the methods described previously [27]. Briefly, CE-TOFMS analysis was conducted using an Agilent CE capillary electrophoresis system equipped with an Agilent 6210 time-of-flight mass spectrometer (Agilent Technologies, Inc., Santa Clara, CA, USA). The systems were controlled via Agilent G2201AA ChemStation software version B.03.01 (Agilent Technologies) and were connected via fused silica capillary (50 μm i.d. × 80 cm total length) with commercial electrophoresis buffer (H3301-1001 and I3302-1023 for cation and anion analyses, respectively, HMT) as the electrolyte. The spectrometer was scanned from m/z 50 to 1,000, and peaks were extracted using the MasterHands (Keio University, Tsuruoka, Yamagata, Japan), an automatic integration software, to obtain the peak information, including m/z, peak area, and migration time (MT) [28]. Signal peaks corresponding to isotopomers, adduct ions, and other product ions of known metabolites were excluded, and the remaining peaks were annotated according to the HMT metabolite database based on their m/z values and MTs. The areas of the annotated peaks were then normalized to internal standards and sample amounts to obtain the relative concentrations of each metabolite.

## Statistical analyses

The data were analyzed using the JMP15 software (SAS Institute Inc., Cary, NC, USA). In addition, the data are hereby expressed as medians with interquartile ranges or as averages ± standard deviations (SDs). The non-parametric Kruskal-Wallis test was conducted to assess the differences between the groups. On the other hand, the $\chi^2$ test was conducted to assess the categorical variables. Spearman's correlation coefficient was used for the correlation analysis. The area under the receiver operating characteristic (AUROC) analysis was performed to confirm the usefulness of predicting an outcome and to assess the usefulness of generating optimal cut-offs based on the Youden Index. The Kaplan-Meier analysis was used to determine the cumulative percentage of survival or novel HCC emergence. The Gehan-Breslow-Wilcoxon was used to compare the differences between the groups. A Cox proportional hazard analysis was done to build a model that was stratified based on clinical presentation for

outcome prediction. In the metabolome analysis, hierarchical cluster analysis (HCA), principal component analysis (PCA), and discriminant analysis using the partial least squares (PLS-DA) method were performed using the HMT's proprietary MATLAB and R programs, respectively. PCA was used to detect outliers and to obtain an overview of the variation among the groups. PLS-DA was applied to cluster observations with similar metabolite profiles and to identify metabolites accounting for discrimination between groups. Welch's t-test was used to analyze the differences between the progression and regression groups. The results were considered significant at $P < 0.05$.

## Results

### Background characteristics and outcomes

The background characteristics of the study participants are presented in Table 1. The median observation period was 48 months (ranges, 6–64 months). Of the 326 patients with CHC aged 70 ± 12 years, 208 patients (63.8%) were female, and 118 were male. A total of 60 patients (18.4%) had a prior history of HCC, and all of their HCC were treated before DAAs. The fact

**Table 1. Background characteristics, clinical parameters, and outcomes of the study participants.**

| Parameters | All patients |
|---|---|
| N | 326 |
| Background parameters | |
| Sex (M/F), N (%) | 118 (36.2) / 208 (63.8) |
| Age, years | 70 ± 12 |
| SOF-based DAAs, N (%) | 210 (64.4) |
| Prior history of HCC, N (%) | 60 (18.4) |
| Liver transplanted, N (%) | 11 (3.4) |
| Non-invasive fibrosis parameters | |
| Pre platelet count, $\times10^4$/μL | 15.9 ± 6.8 |
| Percentage of Pre platelet count $< 10^5$/μL | 19.7% |
| Pre FIB-4 index | 4.7 ± 6.6 |
| Percentage of Pre FIB-4 $> 3.25$ | 49.2% |
| Pre APRI | 1.5 ± 3.6 |
| Percentage of Pre APRI $> 2.0$ | 17.6% |
| Pre Forns index | 7.3 ± 2.2 |
| Percentage of Pre Forns index $< 4.2$ | 7.4% |
| Pre 4COL7S, ng/mL | 6.7 ± 2.6 |
| Percentage of Pre 4COL7S $\geq 6.1$ ng/mL | 50.0% |
| Clinical Outcomes | |
| HCC emerged post SVR12, N (%) | 40 (12.3) |
| All-cause mortality, N (%) | 10 (3.1) |
| 4COL7S-defined fibrosis progression$_{SVR}$[†], N (%) | 97 (29.8) |
| 4COL7S-defined fibrosis regression$_{SVR}$[†], N (%) | 229 (70.2) |

Data are shown as mean ± standard deviation.

[†] 4COL7S-defined fibrosis progression$_{SVR}$ or regression$_{SVR}$: a change rate of serum 4COL7S from pre-treatment to SVR12 $\geq$ 0% is defined as fibrosis progression SVR; and $< 0\%$ is defined as fibrosis regression SVR.

Abbreviations: M, male; F, female; SOF, sofosbuvir; DAAs, direct antiviral agents; Pre, pre-treatment; 4COL7S, type IV collagen 7S fragment; HCC, hepatocellular carcinoma; SVR12, sustained virological response at week 12 after end of treatment.

that no recurrence of HCC was confirmed before the start of DAAs for at least 3 months. Eleven patients (3.4%) were liver transplanted (Table 1). None of the patients suffered from any admission for hepatic decompensation, including over symptoms of portal hypertension, esophageal and gastric variceal bleeding, or onset of ascites. The study subjects were comprised of 49.2% of patients with FIB-4-defined significant fibrosis ($\geq$ 3.25; over F3) and 17.6% of patients with APRI-defined cirrhosis ($\geq$ 2.0) at baseline. Ten patients (3.1%) died during observation. Their causes of death are summarized in S1 Table in S1 File. Notably, liver-related mortality due to HCC was observed in three of them (30%), while non-liver-related mortality was observed in seven patients (70%). We observed 40 cases (12.3%) with at least one episode of new HCC emergence after SVR12 (Table 1).

At baseline, serum levels of 4COL7S were significantly correlated with clinical fibrosis parameters, including peripheral platelet counts, FIB-4, APRI, and Forns indices (S2A-S2D Fig in S1 File). Patients with FIB-4 indices > 3.25 had significantly higher 4COL7S than those without (S2E Fig in S1 File). A serum level of 4COL7S$\geq$6.1 ng/mL could predict an FIB-4 index>3.25 with an AUROC of 0.86 (S2F Fig in S1 File; sensitivity, 82.80%; specificity, 81.48%; positive predictive value, 81.25%; negative predictive value, 83.02%). Therefore, we used these cutoff values to define advanced fibrosis. Baseline degree of liver fibrosis at a cut-off of 4COL7S$\geq$6.1 ng/mL significantly associated with all-cause mortality (S2G Fig in S1 File; $P$ = 0.0155) and newly emerged HCC (S2H Fig in S1 File; $P$< 0.01) after SVR.

## Dynamics of serum 4COL7S until SVR12 in DAA-treated patients with CHC

After IFN-free DAA treatment, we observed a significant global decrease in serum 4COL7S levels between pre and EOT or SVR12 (Fig 1A). However, we observed two patterns in serum 4COL7S levels (Fig 1B). Patients who had higher levels of serum 4COL7S at SVR12 were classified into the 4COL7S-defined fibrosis progression group at SVR12 (Progression$_{SVR}$ or change rate of 4COL7S [$\Delta$%4COL7S$_{SVR-Pre}$]$\geq$ 0). Patients with decreased serum levels of 4COL7S were classified into the 4COL7S-defined fibrosis regression group at SVR12 (Regression$_{SVR}$; or change rate of 4COL7S [$\Delta$%4COL7S$_{SVR-Pre}$]< 0). The Progression$_{SVR}$ group consisted of 29.8% of the study subjects (Table 1; inclusive of 10 patients [3.1%], with change rates of 0%). Notably, patients with fibrosis regression$_{SVR}$ had higher background liver fibrosis levels (higher baseline 4COL7S in Fig 1B, and other clinical parameters in S2 Table in S1 File) than those with progression$_{SVR}$. No other background factors differed significantly between them.

## 4COL7S-defined fibrosis progression at SVR12 predicted post-SVR all-cause mortality

We analyzed whether 4COL7S-defined fibrosis progression and regression at SVR12 predicted clinical outcomes. In the Kaplan-Meier analyses, patients with 4COL7S-defined fibrosis progression$_{SVR}$ had significantly higher all-cause mortality than those with regression$_{SVR}$. Their mortality rates were 3.16% and 0% at 12 months after SVR12, and 6.73% and 0% at 24 months (Fig 2A, $P$ = 0.0062), respectively. However, the probability of newly emerged HCC did not differ significantly between the two groups (Fig 2B; $P$ = 0.24). In a sub-analysis of patients with pre-treatment serum 4COL7S$\geq$ 6.1 ng/mL, which was suggestive of advanced fibrosis (n = 163), the tendency for 4COL7S-defined fibrosis progression SVR to predict all-cause mortality, but not newly emerged HCC became even more prominent (Fig 2C and 2D). This phenomenon was also observed in the other analyses when study subjects were stratified based on FIB-4 indices > 3.25, which suggested advanced fibrosis (n = 157) (S3A, S3B Fig in S1 File), or in patients with a prior history of HCC (n = 60, S3C, S3D Fig in S1 File) or in those without

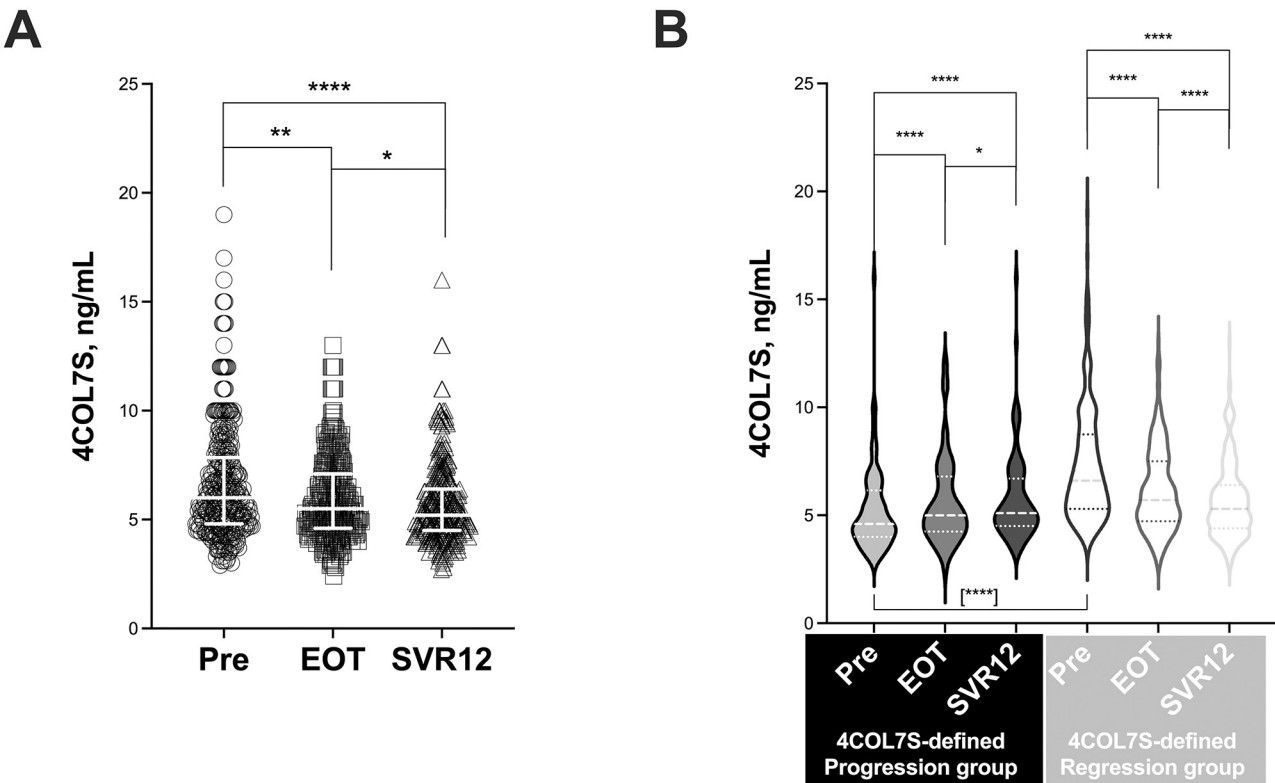

**Fig 1. Peri-treatment dynamics of serum type IV collagen fragment 7S.** (A) Serum levels of type IV collagen fragment 7S (4COL7S; ng/mL) just before IFN-free DAAs (Pre, circles), at the end of treatment (EOT, squares), and at sustained virological response at 12 weeks after the end of treatment (SVR12, triangles) are shown. (B) Serum levels of 4COL7S at the three timepoints in the 4COL7S-defined fibrosis progression$_{SVR}$ and regression$_{SVR}$ groups are shown as violin plots. Data are presented as median with interquartile ranges. Statistical difference between baseline levels of 4COL7S of the two groups is shown in brackets. *, $P < 0.05$; **, $P < 0.01$; ****, $P < 0.0001$.

(n = 266, S3E, S3F Fig in S1 File). We did not observe any patients suffering from hepatic decompensation. With regard to liver-related mortality, patients with 4COL7S-defined fibrosis progression SVR tended to have higher liver-related mortality than those with SVR ($P = 0.09$, S4A Fig in S1 File). Again, in a sub-analysis of patients with pre-treatment serum 4COL7S≥ 6.1 ng/mL, which was suggestive of advanced fibrosis (n = 163), 4COL7S-defined fibrosis progression$_{SVR}$ was significantly associated with liver-related mortality ($P = 0.0019$, S4B Fig in S1 File).

Among patients with advanced fibrosis (n = 163 with nine censored all-cause death), we conducted a Cox proportional hazard analysis based on age ($> 60$ years), sex (male), prior history of HCC, pre-treatment platelet count ($< 10x\ 10^4/\mu L$; suggestive of baseline advanced fibrosis), serum albumin at SVR, estimated glomerular filtration rate (eGFR) at SVR, and 4COL7-defined fibrosis progression SVR. Only 4COL7S-defined fibrosis progression$_{SVR}$ (multivariate $P = 0.0051$) and pre-treatment platelet count (multivariate $P = 0.0308$) significantly predicted all-cause mortality (Table 2).

## Exploratory analysis 1: Dynamics of 4COL7S and serum MMPs/TIMP-1

We conducted a sub-cohort analysis (exploratory analysis 1) in 23 patients whose serum MMP2, MMP9, and TIMP-1 levels were available. The background characteristics of these 23 patients are summarized in S3 Table in S1 File. In the analysis of the whole cohort, we showed

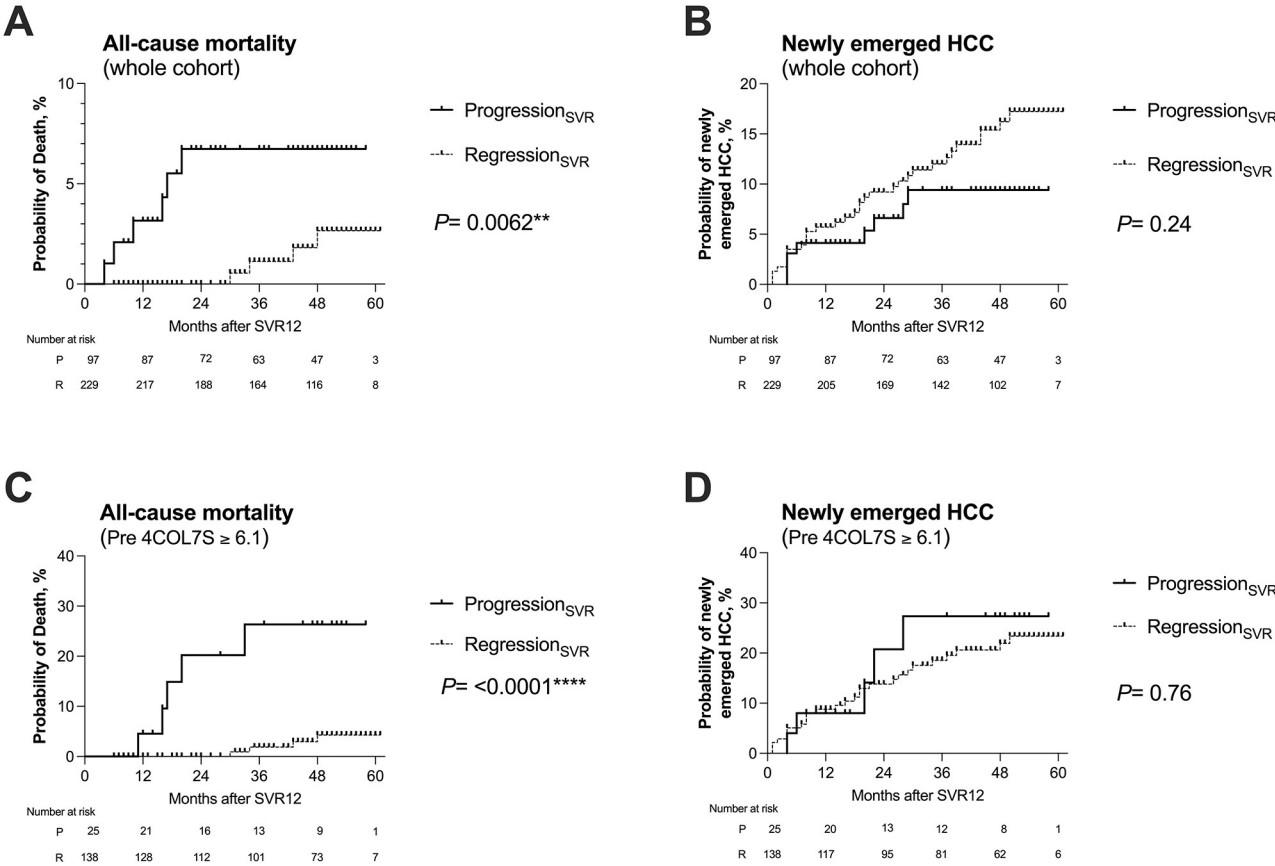

**Fig 2. Kaplan-Meier analyses of all-cause mortality and the new emergence of HCC post-SVR stratified by 4COL7S-defined fibrosis progression$_{SVR}$ and regression$_{SVR}$.** A comparison of 4COL7S-defined fibrosis progression$_{SVR}$ and regression$_{SVR}$ (refer to the main text for definition), all-cause mortality (in panel A), and cumulative frequency of newly emerged HCC (in panel B) is evaluated from SVR12 in 326 IFN-free DAA-treated patients achieving SVR (whole cohort). All-cause mortality (in panel C) and accumulative frequency of newly emerged HCC (in panel D) are evaluated based on the SVR12 of patients with baseline serum levels of 4COL7S≥ 6.1 ng/mL (n = 163). **P < 0.01; ****P <0.0001.

**Table 2. Cox proportional hazard model for factors associated with all-cause mortality in patients whose baseline serum 4COL7S ≥ 6.1 ng/mL (N = 163 with nine censored events).**

| Variables | P (uni) | Hazard ratio (95% CI) | P (multi) |
|---|---|---|---|
| Age > 60 years | 0.62 | 1.25 (0.22–7.21) | 0.80 |
| Sex, male | 0.73 | 2.13 (0.45–10.1) | 0.34 |
| Prior history of HCC, yes | 0.45 | 1.29 (0.41–12.5) | 0.73 |
| Platelet count < 10 × 10$^4$/μL, pre-treatment | 0.0271* | 6.62 (1.19–36.9) | 0.0308* |
| Serum albumin < 3.5 g/dL, at SVR12 | 0.12 | 5.31 (0.88–31.8) | 0.07 |
| eGFR < 60 mL/min/1.73m$^2$, at SVR12 | 0.49 | 2.27 (0.41–12.5) | 0.35 |
| 4COL7S-defined fibrosis Progression$_{SVR}$[†] | 0.0141* | 9.12 (2.34–35.4) | 0.0051** |

Factors that are statistically not significant by univariate analyses: pre-treatment FIB-4 index (P = 0.92), APRI (P = 0.62), and Forns index (P = 0.19).

[†] 4COL7S-defined fibrosis progression$_{SVR}$: a change rate of serum 4COL7S from pre-treatment to SVR12 ≥ 0% is defined as fibrosis progression$_{SVR}$.

Abbreviations: 4COL7S, type IV collagen 7S fragment; uni, univariate; HCC, hepatocellular carcinoma; SVR, sustained virological response; eGFR, estimated glomerular filtration rate

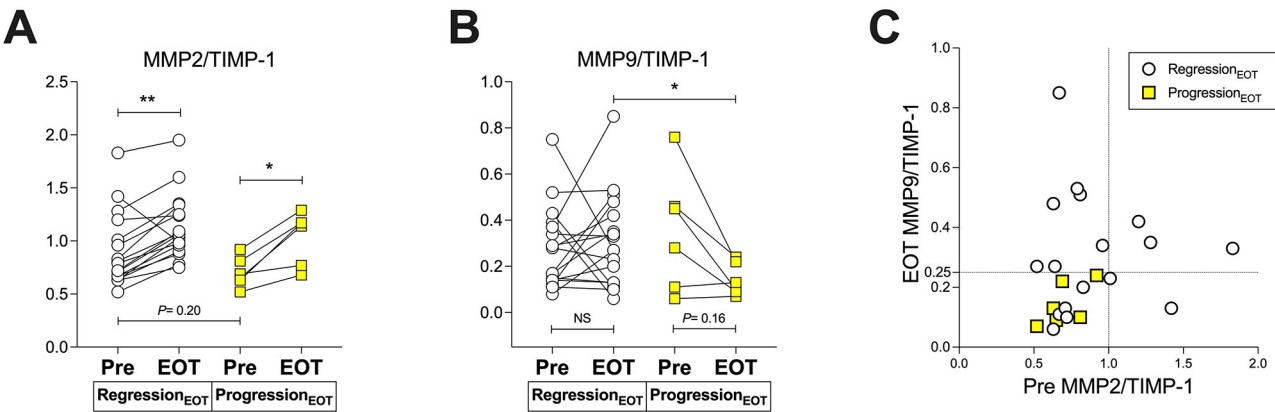

**Fig 3. Associations between the serum dynamics of 4COL7S and MMP and TIMP-1 ratios from pre-treatment to end of treatment (exploratory analysis 1).** Of 23 IFN-free DAA-treated patients achieving SVR, serum MMP2/TIMP-1 ratios at pre and EOT (in panel A) and serum MMP9/TIMP-1 ratios at pre and EOT (in panel B) are shown comparing 4COL7S-defined fibrosis progression$_{EOT}$ (yellow squares) and regression$_{EOT}$ (clear circles). (C) The relationships of pre-treatment MMP2/TIMP-1 and MMP9/TIMP-1 at EOT presented with 4COL7S-defined fibrosis progression (yellow squares) and regression (clear circles) at EOT are shown. Data are presented as medians with interquartile ranges. *$P < 0.05$.

that the change rates of 4COL7S at EOT were significantly and highly correlated with those at SVR12 (S5A Fig in S1 File), and 4COL7S-defined fibrosis progression SVR could be predicted by 4COL7S-defined fibrosis progression at EOT (e.g., change rates of 4COL7S at EOT or progression$_{EOT}$, S5B Fig in S1 File). Correlations between serum 4COL7S and MMP2, MMP9, and TIMP-1 at pre-treatment or at EOT are shown in S6 Fig in S1 File. Interestingly, a tendency for higher pre-treatment MMP2/TIMP-1 and significantly higher MMP9/TIMP-1 at EOT were associated with regression$_{EOT}$ (Fig 3A and 3B). In addition, a global increase in MMP2/TIMP-1 was observed in each group (Fig 3A), while MMP9/TIMP-1 was relatively maintained in the regression$_{EOT}$, but not in the progression$_{EOT}$ (Fig 3B). Values of pre-treatment MMP2/TIMP-1 >1.0 or MMP9/TIMP-1 >0.25 at EOT were used to predict regression$_{EOT}$ with a sensitivity of 71% and a specificity of 100% (Fig 3C). Since higher MMPs-to-TIMP-1 ratios suggest fibrolysis, the peri-treatment decrease of 4COL7S reflects a matrix-degrading phenotype after HCV eradication.

## Exploratory analysis 2: Metabolomic correlations

To determine whether metabolomic changes characterized the dynamics of 4COL7S during HCV eradication with DAAs, seven patients with 4COL7S-defined fibrosis progression at EOT and another group of background characteristics-matched seven patients with fibrosis regression were included in a nested case-control study (exploratory analysis 2) for metabolomic changes in paired serum samples using the CE-TOFMS approach. Their clinical backgrounds are shown in S4 Table in S1 File. No significant baseline differences, including age, sex, prior HCC history, 4COL7S levels, FIB-4 indices, and renal function were observed between the two groups. Compared to the pre-treatment levels, the serum 4COL7S was significantly changed in the progression and regression groups, respectively, at EOT (Fig 4A). Among the 185 small-molecule metabolites detected (113 cations and 72 anions), 176 were identified as known molecules. The HCA results for these metabolic patterns are shown in S7 Fig in S1 File. An overview of the samples is shown in the PCA plot. No samples were omitted from further analysis (Fig 4B, left). By using group information in PLS-DA, systematic differences could be seen between the four groups (Fig 4B), demonstrating that similar metabolite profiles might exist within each group, and some might account for discrimination between

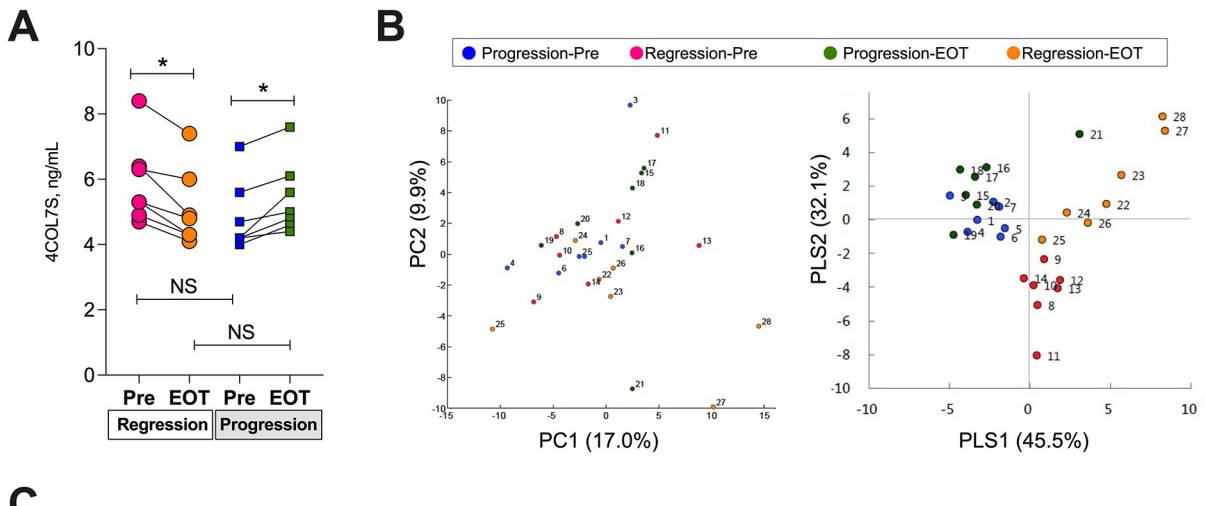

**Fig 4. Metabolomic analysis using CE-TOFMS in patients with 4COL7S-defined fibrosis progression or regression from pre-treatment to end of treatment (exploratory analysis 2).** (A) Seven patients with 4COL7S-defined fibrosis progression at the EOT and another background characteristic-matched seven patients with fibrosis regression are analyzed. Serum levels of 4COL7S before treatment and at EOT are shown. (B) A principal component analysis (PCA) plot (left) and plot generated from a discriminant analysis of a partial least square method (PLS-DA; right) derived from metabolomic analysis using CE-TOFMS of paired serum samples before treatment (Pre) and at EOT of both groups are shown. Each number label indicates an individual sample. (C) Twenty-seven metabolites demonstrating tendencies of differences ($P \leq 0.10$) or substantial ratios of differences (relative concentration ratio $>2.0$) between fibrosis progression and regression groups at Pre or at EOT are shown. Data are shown as averages with standard deviations. $^*P < 0.05$, $^{**}P < 0.01$. NS, not significant; ND, not detected.

the peri-treatment dynamics of 4COL7S. A total of 27 metabolites demonstrating differences ($P \leq 0.10$) or substantial ratios of differences (relative concentration ratio > 2.0) between fibrosis progression and regression groups at pre-treatment or at EOT are shown in Fig 4C. A richness in relative concentration favoring 4COL7S-defined fibrosis regression of butyrate, a short-chain fatty acid (SCFA), and medium-chain fatty acids (hexanoic acid and pelargonic acid), carbohydrate conjugates such as glycerophosphocholine (GPC), anabolic amino acids (glutamine, proline, and tryptophan), carnitine, adenosine, and ascorbate 2-sulfate (a metabolite of vitamin C, a cofactor in at least eight enzymatic reactions including several collagen synthesis reactions) were observed. On the other hand, a richness in relative concentration favoring fibrosis progression of taurocholic acid (TCA), a conjugated primary bile acid, and various medium-chain fatty acids, fatty acid conjugates, cysteine glutathione disulfide (formed upon oxidative stress of glutathione), urea, and uremic-related toxins (guanidinosuccinic acid [29] and huppuric acid [30]) were observed.

## Discussion

In this observational study of 326 patients with CHC whose HCV was eradicated using IFN-free DAAs, we demonstrated that the peri-treatment dynamics of serum 4COL7S, a non-invasive fibrosis marker, significantly correlated with all-cause mortality.

All-cause mortality after SVR has been highlighted in several studies. A prospective observational study from France [31] and various retrospective database analyses [32, 33] revealed that SVR was associated with an improved all-cause mortality in patients with CHC. However, extrahepatic benefits from HCV eradication were less prominent than the reduction in liver-related mortality [33]. Another large-scale database analysis demonstrated that after the widespread use of DAAs, mortality from cardiovascular disease, extrahepatic cancers, or type 2 diabetes mellitus (T2DM) was increased only among patients with HCV infection but not in those with other CLDs such as HBV, alcoholic liver disease, and nonalcoholic fatty liver disease [34]. Therefore, the prediction of non-liver-related mortality is an unmet need in the post-DAA era. Various extrahepatic factors, such as low BMI [32], decreased eGFR [32], hypertension [32] and co-infection with HIV [35], have been reported to negatively impact overall survival in patients with CHC after SVR. As demonstrated in S2G, S2H Fig in S1 File, CHC patients with baseline advanced fibrosis suffered from higher all-cause mortality and frequency of *de novo* HCC after SVR than those without. Moreover, in the current study, 4COL7S-defined fibrosis progression$_{SVR}$ significantly predicted all-cause, preferentially non-liver-related mortality in patients with CHC after HCV eradication (Fig 2A and S4A Fig in S1 File). In addition, the Cox proportional hazard model suggested that aside from the baseline levels of liver fibrosis (platelet counts), the peri-treatment dynamics of ECM remodeling (4COL7S-defined fibrosis progression$_{SVR}$) might be independent prognostic factors in CHC patients with advanced fibrosis after SVR (Table 2).

A few previous reports have also suggested that non-invasive liver fibrosis tests at SVR might be a key prognostic factor for all-cause mortality in patients with CHC after HCV eradication. In an observational study involving 640 HIV-coinfected patients with CHC and advanced fibrosis treated with DAAs conducted in Span, Corma-Gomez et al. demonstrated that liver stiffness assessed via transient elastography at SVR was the only significant predictive factor for all-cause mortality [36]. In another observational study involving 947 patients with CHC without a history of HCC treated with DAAs conducted in Japan, Nakagawa et al. showed that higher levels of Mac-2 binding protein glycosylation isomer cutoff indices (a serological liver fibrosis marker) at SVR were used to predict worse overall survival [37]. However, in those previous reports, whether the status of liver fibrosis at SVR or a driving force for ECM

remodeling toward fibrosis resolution during treatment with DAA is more influential for overall survival was not discussed. The results of our current study suggest that both may play important roles.

Whether the dynamics of non-invasive fibrosis markers reflect real intrahepatic ECM mass after SVR and the best way to evaluate liver fibrosis reversion after viral suppression or eradication remain inconclusive [13, 38]. Non-invasive fibrosis markers may be influenced by both reduced intrahepatic inflammation and by reduced systemic inflammation following viral eradication. Thus, the latest European Association of the Study of the Liver clinical practice guidelines state that non-invasive scores and liver stiffness by various elastography methods are not accurate in detecting fibrosis regression after SVR in HCV patients diagnosed with compensated advanced chronic liver diseases after antiviral therapy [38]. Although non-invasive serological fibrosis markers may not be as robust as histologic assessment, trends toward improvement of these markers suggestive of fibrosis reversal have been shown in previous reports [10, 11]. In the exploratory analysis involving one of the 23 patients with CHC treated with IFN-free DAAs, we demonstrated the dynamics of serum 4COL7S at EOT correlated with individual serum MMP to TIMP-1 ratios (Fig 3). A principal feature of hepatic fibrosis is the imbalance between MMPs and TIMPs. Both MMPs and TIMPs are indispensable in fibrogenesis and fibrolysis. MMP2 (gelatinase A) and TIMP-1 are produced mainly by activated HSCs during hepatic fibrogenesis [16, 39]. MMP9 (gelatinase B) is produced by many parenchymal and non-parenchymal cells [40]. Possible chronological expression profiles with a constituently elevated expression of MMP2 from fibrogenesis to later phases of fibrolysis after cessation of persistent liver injury and an abrupt elevation in MMP9 expression during the early phase of fibrolysis have been considered [41]. The fact that 4COL7S-defined fibrosis regression at EOT was correlated with MMP2/TIMP-1 at pre-treatment and MMP9/TIMP-1 at EOT, as we demonstrated in this study (Fig 3), was congruent with such chronological expression profiles.

Another important issue is whether changes in non-invasive markers are correlated with clinical outcomes. A large observational study demonstrated that using the same cut-off value of 3.25, high FIB-4 scores at SVR were associated with an increased risk for HCC [42], even though intrahepatic fibrosis progression may not be exactly defined. 4COL7S, rather than a marker of collagen degradation, is considered a marker of liver fibrogenesis [18]. Elevated levels of type IV collagen, the major constituent of the basement membrane that is widespread throughout tissues, have been reported to be associated with various extrahepatic pathological states such as atherosclerosis [43], acute heart decompensation [44], chronic obstructive pulmonary disease [45], interstitial lung diseases [46], and chronic kidney disease [47]. Moreover, significant diagnostic superiority of 4COL7S to FIB-4, APRI, and NAFLD fibrosis scores has been reported to identify advanced fibrosis in NAFLD in patients with T2DM but not in those without [22]. Both atherosclerosis-related cardiovascular diseases and T2DM are well-known extrahepatic manifestations of CHC [5]. An epidemiological survey revealed that there was an increase in cardiovascular and T2DM-realted mortality in patients with CHC but not in those with other CLDs in the post-DAA era [34]. Not to mention that post-SVR fatty liver [48] and comorbidity of T2DM [49] are also reported to be factors for increased hepatocarcinogenesis in patients with CHC after HCV eradication with IFN-based therapy. Since many of the "non-hepatic" causes reflected by serum 4COL7S levels are also possible prognostic factors for all-cause mortality in patients with CHC after SVR, this may be the reason why 4COL7S correlates with survival prognosis in these patients.

In exploratory analysis 2 of a nested case-control metabolomic analysis, we revealed that the peri-treatment dynamics of 4COL7S were prominently correlated with metabolomic alterations after HCV eradication, which also supported a possible link between its prognostic impact and extra-hepatic factors. As shown in Fig 4C, fatty acid metabolism has been shown to

be a cardinal hallmark of chronic HCV infection [14]. Moreover, some metabolites have been focused on in CLD in previous reports. For instance, TCA has been reported to be HSC-activating and pro-fibrogenic [50]. Depletion of GPC has been associated with sarcopenia, predicting allograft failure after liver transplantation [51]. Interestingly, even without obvious differences in eGFR (S4 Table in S1 File), increased uremic toxin levels associated with peri-treatment increased 4COL7S levels (Fig 4C), implicating a possible liver-kidney association during liver fibrosis remodeling. In addition, Ponziani et al. reported that HCV eradication with DAAs was associated with a modification of the gut microbiota in patients with cirrhosis [52]. As we demonstrated significant changes in butyrate, a SCFA, and TCA, a conjugated primary bile acid, the gut-liver axis might also play a presently unraveled role in ECM remodeling and host prognosis post-SVR.

This study has some limitations. Histological assessment for reversion of liver fibrosis and data of serial evaluation for transient elastography after HCV eradication were not available in this study; therefore, the associations between 4COL7S were unproven. The number of observed events in patients with less advanced liver fibrosis (serum 4COL7S less than 6.0 ng/mL before treatment) was small. Therefore, generalizability in patients with less advanced fibrosis may be limited, and further observations with larger numbers of study subjects are needed. Underlying diseases that may also influence ECM remodeling in other organs may cause possible bias in the assessment of the dynamics of 4COL7S. The influence of DAAs themselves on the dynamics of collagen turnover and metabolism is also unknown; the inclusion of various kinds of IFN-free DAAs in this study may improve generalizability of this study but may also increase inhomogeneity. Although admission due to hepatic decompensation was not observed, the information of clinical features of portal hypertension were not included in the analysis. The numbers of both exploratory analyses were still small, and these results need to be verified in studies with larger numbers. How a biomarker for prediction of all-cause mortality can be adequately utilized in clinical settings remains to be fu determined.

To conclude, we provide evidence that peri-treatment dynamics of 4COL7S, a serum fibrosis marker, predict all-cause mortality post-SVR. They may also help in risk stratification after HCV eradication. CHC and HCV eradication, especially those linking liver fibrosis reversion and metabolism, and clinical outcomes may help refine our knowledge and improve the management of advanced fibrotic liver diseases in the future.

## Supporting information

**S1 File.**
(PDF)

**S1 Data.**
(XLSX)

**S1 Checklist.**
(DOC)

## Author Contributions

**Conceptualization:** Karin Yamataka, Po-sung Chu, Hirotoshi Ebinuma, Takanori Kanai, Nobuhiro Nakamoto.

**Data curation:** Karin Yamataka, Po-sung Chu, Yuzo Koda, Nobuhito Taniki, Rei Morikawa, Aya Yoshida, Fumie Noguchi, Ryosuke Kasuga, Takaya Tabuchi, Hirotoshi Ebinuma.

**Formal analysis:** Karin Yamataka, Po-sung Chu, Yuzo Koda, Nobuhito Taniki, Rei Morikawa, Aya Yoshida, Fumie Noguchi, Ryosuke Kasuga, Takaya Tabuchi, Hirotoshi Ebinuma.

**Funding acquisition:** Po-sung Chu, Takanori Kanai.

**Investigation:** Karin Yamataka, Po-sung Chu, Yuzo Koda, Nobuhito Taniki, Rei Morikawa, Aya Yoshida, Fumie Noguchi, Ryosuke Kasuga, Takaya Tabuchi, Hirotoshi Ebinuma, Nobuhiro Nakamoto.

**Methodology:** Po-sung Chu, Yuzo Koda, Nobuhito Taniki, Rei Morikawa, Ryosuke Kasuga, Hirotoshi Ebinuma, Nobuhiro Nakamoto.

**Project administration:** Po-sung Chu, Hirotoshi Ebinuma, Takanori Kanai, Nobuhiro Nakamoto.

**Resources:** Karin Yamataka, Po-sung Chu, Yuzo Koda, Nobuhito Taniki, Rei Morikawa, Aya Yoshida, Fumie Noguchi, Takaya Tabuchi, Takanori Kanai, Nobuhiro Nakamoto.

**Software:** Karin Yamataka, Po-sung Chu, Yuzo Koda.

**Supervision:** Hirotoshi Ebinuma, Takanori Kanai, Nobuhiro Nakamoto.

**Validation:** Po-sung Chu, Aya Yoshida, Ryosuke Kasuga, Takaya Tabuchi.

**Visualization:** Karin Yamataka, Po-sung Chu.

**Writing – original draft:** Karin Yamataka, Po-sung Chu.

**Writing – review & editing:** Hirotoshi Ebinuma, Takanori Kanai, Nobuhiro Nakamoto.

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
