## [Decision Letter · Decision Letter 0]

29 Aug 2022

PONE-D-22-21128Dynamics of type IV collagen 7S fragment on eradication of HCV with direct antiviral agents: prognostic and metabolomic impactsPLOS ONE

Dear Dr. Chu,

Thank you for submitting your manuscript to PLOS ONE. After careful consideration, we feel that it has merit but does not fully meet PLOS ONE’s publication criteria as it currently stands. Therefore, we invite you to submit a revised version of the manuscript that addresses the points raised during the review process.

In the discussion, it needs to add a more explicit explanation about the underlying mechanisms to the current findings to improve the scientific merits which is a critical criterion for future acceptance of the paper.Please ensure that your decision is justified on PLOS ONE’s publication criteria and not, for example, on novelty or perceived impact.

We look forward to receiving your revised manuscript.

Kind regards,

Youkyung H. Choi, Ph.D.

Academic Editor

PLOS ONE

Journal Requirements:

2. You indicated that you had ethical approval for your study. Please clarify whether minors (patients below the age of 18 years) were included in your study. If yes, in your Methods section, please ensure you have also stated whether you obtained consent from parents or guardians of the minors included in the study or whether the research ethics committee or IRB specifically waived the need for their consent.

"This study was supported in part by grants-in-aid from the Keio Gijuku Academic Development Funds and the Mitsubishi Tanabe Pharma Corporation. The funders had no role in study design, data collection and analysis, decision to publish, or preparation of the manuscript".

"Y.K. is an employee at Mitsubishi Tanabe Pharma Corporation. The remaining authors declare no competing interests". 

Additional Editor Comments:

Dear Dr. Chu,

Thank you for submitting your manuscript to PLOS ONE. After careful consideration, we feel that it has merit, but is not suitable for publication as it currently stands. Therefore, my decision is "Major Revision."

We invite you to submit a revised version of the manuscript that addresses all points below by the reviewers. Particularly, a more explicit explanation about the underlying mechanisms to the current findings to improve the scientific merits which is a critical criterion for future acceptance of the paper.

Reviewers' comments:

Reviewer's Responses to Questions

**Comments to the Author**

1. Is the manuscript technically sound, and do the data support the conclusions?

Reviewer #1: Yes

Reviewer #2: Yes

2. Has the statistical analysis been performed appropriately and rigorously? 

Reviewer #1: Yes

Reviewer #2: Yes

3. Have the authors made all data underlying the findings in their manuscript fully available?

Reviewer #1: Yes

Reviewer #2: Yes

4. Is the manuscript presented in an intelligible fashion and written in standard English?

Reviewer #1: Yes

Reviewer #2: Yes

5. Review Comments to the Author

Reviewer #1: The authors assessed the dynamics of 4COL7S in HCV patients who attained SVR with DAAs and correlated the trends of 4COL7S with the all-cause mortality and liver-related mortality. Furthermore, they also performed exploratory studies to prove the association of fibrinolysis/fibrogenesis markers and metalomic profiles with 4COL7S dynamics to corroborate the systemic and intra-hepatic change of biomarkers. The manuscript is interesting and can provide insight as the clinical utility for the outcome prediction in HCV-SVR patients.

1. Supplementary Figure 1: The range of symbol of changes of 4COL7S should be extended from EOT to SVR12 because the authors stated that the changes were calculated by 4COL7S values at SVR12 minus those at baseline in Table 1 footnotes. Please confirm the time interval of the 4COL7S-defined progression or regression in the current study (baseline to EOT or baseline to SVR12).

2. Line 157: Please state clearly the kit information used to determined SVR (limit of detection, manufacturer, company, kits name, countries etc…). Furthermore, please also provide the similar information about 4COL7S (line 196).

3. Line 173: The description “Before, during, and after achieving SVR” was confusing, and please re-write this sentence.

4. Line 183: Table 1 should be moved to the Results section. Furthermore, please trim the wordings in line 183.

5. Line 290: I guessed that cut-off value of 6.1 was chosen by Youden index, and please state the sensitivity, specificity, PPV and NPV to predict FIB-4 > 3.25 with this cut-off value.

6. It is intriguing why patients with more severe hepatic fibrosis at baseline tended to have regression of 4COL7S, while those with milder ones had the opposite trend. Please explain and proposed the potential mechanism.

7. It is also surprising to know that as high as 30% of patients who achieved SVR12 still had fibrosis progression based on the dynamics of 4COL7S, compared to the published reports using dynamic changes of FIB-4, transient elastography, etc… Could the authors have some explanation for this phenomenon?

8. Although ACOL7S, MMP and TIMP are indicators of fibrogenesis and fibrinolysis, they are not specific for the liver. While patients with HCV who achieve SVR may reduce intrahepatic inflammation which favors fibrinolysis, viral eradication also reduces the risk of systemic inflammation that also favors reduction of systemic fibrogenesis, with which noninvasive fibrosis markers are expected to be improved. That means as I proposed in Q6 and Q7 as how the authors explain a high percentage of progression by 4COL7S dynamics in SVR patients.

9. Lines 332 and 340: The authors stated that a baseline 4COL7S > 6.1 had the similar trend of all-cause or liver-related mortality and HCC with regression or progression of 4COL7S to overall population. How about the trends in patients with a baseline 4COL7S < 6.1? It would be important because the similar or different trends in patients with > 6.1 or > 6.1 may imply that the baseline advanced hepatic fibrosis would be a disease modifier or just a confounder.

10. Line 491: “640 HCV-coinfected patients with CHC” should be “HIV-coinfected”.

Reviewer #2: The goal of this study is to find non-invasive fibrosis markers and study their capability to predict post-SVR prognosis in patients. They analyzed the predictive value of Type IV collagen 7S after SVR in addition to trying to identify novel targets that can serve as non-invasive markers for liver fibrosis and post-SVR prognosis in patients.

Major suggestions:

1. The test used for serum 4COL7S level measurement should be more specifically described in the materials and methods. Since kits are commercially available full information about the kits should be provided. Is this kit clinically used specifically for liver fibrosis or is it approved for another disease? Are there any scientific studies that previously evaluated this kit? They should be cited. (lines 196-199)

2. In the metabolome section, the filtrate was evaporated and then reconstituted (lines 221-223). Why was this step required? Were the samples stored and transported in this state?

3. The utility of the 4COL7S test should be better described by the authors. Some of the questions that should be answered are: a) Is there any value in using this test for prognosis of a single patient? b) What if this patient did not have this test done before the treatment? Is there still a use for it if tested post SVR? c) Although a difference between the groups is statistically significant, there is a large overlap between the groups. Is there a plan to study specificity and sensitivity of this test in predicting disease outcome?

4. The metabolomic analysis is based on the 4COL7S-defined fibrosis. Why was it not based on the currently used clinical evaluation of the patients.

5. What are the plans for the metabolomic analysis results? What was the purpose of performing this experiment, and how will these results used in the future.

Minor suggestions:

1. This manuscript uses a lot of abbreviations, and although they are all defined in the beginning of the manuscript, it gets difficult to follow and clearly understand some parts of this manuscript.

2. In addition, readability may be improved by having a native English speaker edit the manuscript.

6. PLOS authors have the option to publish the peer review history of their article (what does this mean?). If published, this will include your full peer review and any attached files.

Reviewer #1: No

Reviewer #2: No

---

## [Author Response · Author response to Decision Letter 0]

4 Oct 2022

Please refer to the attached file for point-by-point response to the reviewers.

---

## [Decision Letter · Decision Letter 1]

17 Oct 2022

Dynamics of type IV collagen 7S fragment on eradication of HCV with direct antiviral agents: prognostic and metabolomic impacts

PONE-D-22-21128R1

Dear Dr. Chu,

We’re pleased to inform you that your manuscript has been judged scientifically suitable for publication and will be formally accepted for publication once it meets all outstanding technical requirements.

Kind regards,

Youkyung H. Choi, Ph.D.

Academic Editor

PLOS ONE

Additional Editor Comments (optional):

Reviewers' comments:

Reviewer's Responses to Questions

**Comments to the Author**

1. If the authors have adequately addressed your comments raised in a previous round of review and you feel that this manuscript is now acceptable for publication, you may indicate that here to bypass the “Comments to the Author” section, enter your conflict of interest statement in the “Confidential to Editor” section, and submit your "Accept" recommendation.

Reviewer #1: All comments have been addressed

Reviewer #2: All comments have been addressed

2. Is the manuscript technically sound, and do the data support the conclusions?

Reviewer #1: Yes

Reviewer #2: Yes

3. Has the statistical analysis been performed appropriately and rigorously? 

Reviewer #1: Yes

Reviewer #2: Yes

4. Have the authors made all data underlying the findings in their manuscript fully available?

Reviewer #1: Yes

Reviewer #2: Yes

5. Is the manuscript presented in an intelligible fashion and written in standard English?

Reviewer #1: Yes

Reviewer #2: Yes

6. Review Comments to the Author

Reviewer #1: (No Response)

Reviewer #2: All comments have been addressed, so no additional comments are needed. This manuscript can be accepted at this point.

7. PLOS authors have the option to publish the peer review history of their article (what does this mean?). If published, this will include your full peer review and any attached files.

Reviewer #1: No

Reviewer #2: No

---

## [Editor Report · Acceptance letter]

19 Oct 2022

PONE-D-22-21128R1 

Dynamics of type IV collagen 7S fragment on eradication of HCV with direct antiviral agents: prognostic and metabolomic impacts 

Dear Dr. Chu:

I'm pleased to inform you that your manuscript has been deemed suitable for publication in PLOS ONE. Congratulations! Your manuscript is now with our production department. 

Kind regards, 

on behalf of

Dr. Youkyung H. Choi 

Academic Editor

PLOS ONE